# A Pilot Study on Structural Changes of Choroidal Vasculature Following Intravitreal Anti-VEGF Injection in Neovascular Age-Related Macular Degeneration: Faricimab vs Ranibizumab

**DOI:** 10.3390/jcm14207257

**Published:** 2025-10-14

**Authors:** Takeyuki Nishiyama, Hiromasa Hirai, Kimie Miyata, Tomo Nishi, Tetsuo Ueda, Satoru Kase

**Affiliations:** Department of Ophthalmology, Nara Medical University, 840 Shijo-Cho, Kashihara 634-8521, Japan; k158380@naramed-u.ac.jp (T.N.);

**Keywords:** neovascular age-related macular degeneration, faricimab, ranibizumab

## Abstract

**Objectives**: This paper aims to explore optical coherence tomography (OCT)-based choroidal vascular changes in patients with neovascular age-related macular degeneration (nAMD) treated with anti-vascular endothelial growth factor (VEGF) agents, faricimab and ranibizumab, in a pilot study. **Methods**: This retrospective pilot cohort study enrolled 28 treatment-naïve nAMD patients who received three consecutive intravitreal anti-VEGF injections at Nara Medical University Hospital. In total, 17 patients (61%) were Type 1 MNV and 11 patients (39%) were Type 2 MNV. Patients were divided into a faricimab group (13 eyes) and a ranibizumab group (15 eyes). The type of macular neovascularization (MNV) and the presence of polyps were recorded. The central choroidal thickness (CCT) and the ratio of luminal area to choroidal area (L/C ratio), derived from binarized OCT images, were measured at baseline after the first and third injections. **Results**: Type 1 MNV was observed in 61% of eyes, with polyps confirmed in 53%. There was no significant difference in best corrected visual acuity (BCVA) for both faricimab and ranibizumab during treatment (*p* = 0.12, 0.94, respectively). After the third injection, a dry macula was achieved in 62% of the faricimab group and 60% of the ranibizumab group. In the ranibizumab group, CCT significantly decreased after the first injection, while no significant change was observed in the faricimab group. Conversely, the L/C ratio significantly decreased in the faricimab group after the third injection (*p* = 0.010). Among faricimab-treated eyes, those with type 1 MNV showed a significantly greater reduction in the L/C ratio compared to type 2 MNV (*p* = 0.017). **Conclusions**: This pilot study suggests that faricimab may exert combined anti-VEGF and Ang-2 effects predominantly on type 1 MNV, potentially leading to vascular constriction. These exploratory findings warrant confirmation in larger studies.

## 1. Introduction

Neovascular age-related macular degeneration (nAMD), characterized by the presence of macular neovascularization (MNV), is a leading cause of severe vision loss among the elderly [1]. MNV is generally classified into Type 1 and Type 2 based on the location of the neovascular complex relative to the retinal pigment epithelium (RPE) [2]. In Type 1 MNV, the neovascular complex is located beneath the RPE, whereas in Type 2 MNV, it is located above the RPE in the subretinal space. Visual impairment in patients with MNV is primarily caused by fluid leakage and hemorrhage, which can ultimately lead to fibrous scarring [3]. Subretinal fluid (SRF) is one of the key indicators used to evaluate disease activity in MNV, reflecting ongoing leakage from the neovascular complex [4]. Anti-vascular endothelial growth factor (anti-VEGF) intravitreal injections are effective in the regression of MNV, and various therapeutic agents are used [3,4]. Faricimab has emerged as a promising option due to its unique dual inhibition of both VEGF and angiopoietin-2 (Ang-2) pathways, potentially offering enhanced efficacy and durability compared to other anti-VEGF drugs [5]. On the other hand, ranibizumab, a Fab fragment of human anti-VEGF monoclonal antibody, is another anti-VEGF drug [6]. Although both drugs have distinct structures and mechanisms, there are still no clear standards for the selective use of each medication.

Optical coherence tomography (OCT) imaging is widely used in the diagnosis and treatment of nAMD. Recently, the ratio of luminal area to choroidal area (L/C ratio, also known as choroidal vascularity index or CVI) has been proposed as an OCT imaging biomarker to quantify the vascular and stromal components of the choroid [7,8]. The L/C ratio has been analyzed in a variety of ocular fundus diseases, including nAMD [9,10,11]. Invernizzi A. et al. demonstrated that the L/C ratio increased when nAMD is activated [12]. However, while the L/C ratio has been expected to predict exudative activity of nAMD [11,13,14,15], its application in evaluating pre- and post-treatment changes with anti-VEGF agents remains largely unexplored. Furthermore, no studies have examined the L/C ratio involved with faricimab, and there are no publications comparing faricimab to other drugs. Therefore, this pilot study aims to explore differences in choroidal structural changes in patients with nAMD treated with faricimab compared with ranibizumab in order to provide preliminary insights for future larger-scale investigations.

## 2. Materials and Methods

This retrospective pilot cohort study included patients with nAMD who visited the Department of Ophthalmology at Nara Medical University Hospital between January 2024 and February 2025. Patients received ophthalmic examinations, such as slit-lamp examinations, fundus photographs, fundus examinations, OCT, fluorescein angiography (FA), and indocyanine green angiography (ICGA). Patients with nAMD who were injected intravitreally either faricimab or ranibizumab were extracted from electronic medical records. By reviewing the medical records, this study explored best-corrected visual acuity (BCVA) and OCT images for patients with nAMD before treatment, 2 weeks after the first anti-VEGF vitreous injection, and 1 month after the third injection (3 months after the first injection). Longitudinal changes during treatment were assessed based on BCVA, central choroidal thickness (CCT), and the L/C ratio. The presence of a history of cardiovascular diseases was also checked by the medical records. OCT images were obtained from SD-OCT (Spectralis, Heidelberg Engineering, Heidelberg, Germany) records. Patients who had received their first treatment at that point and received a single drug administered three times consecutively at one-month intervals thereafter were enrolled.

This study excluded patients whose medication was changed midway, those who received an injection only once, and those who had already started treatment at another hospital. Additionally, this study further excluded patients with unclear OCT images, severe myopia (-9D or higher), glaucoma, diabetic retinopathy, active uveitis, or other ocular diseases. From fundus photographs, we confirmed the presence of drusen. This study also confirmed the presence of intraretinal fluid (IRF) or SRF from OCT images. The types of MNV and the presence or absence of polyps were determined based on FA, ICGA, and OCT images. Based on the recent criteria, type 1 MNV was defined as neovascularization originating from the choroid beneath the RPE, while type 2 MNV was defined as neovascularization that extends beyond the RPE into the subretinal space [16]. CCT was measured using the built-in scale bar of the OCT device. Based on previous studies, horizontal OCT scans were binarized using image analysis software (ImageJ, version 1.54 g, Java 1.8.0_345 [64-bit], National Institutes of Health, Bethesda, MD, USA) to calculate the L/C ratio (Figure 1) [7,13].

More precisely, the total choroidal area (TCA) within the vascular arcade was manually selected. The selected region was then binarized using the Niblack thresholding method, which distinguishes luminal (black) and interstitial (white) components. The stromal choroidal area (SCA) was identified and measured. TCA was also calculated. The luminal choroidal area (LCA) was obtained by subtracting SCA from TCA. Finally, the L/C ratio was defined as the percentage of LCA relative to TCA.

Statistical analysis was performed using EZR software (version 1.68, Saitama Medical Center, Jichi Medical University, Saitama, Japan) [17]. EZR is a graphical user interface for R (R Foundation for Statistical Computing, Vienna, Austria). Graph creation was performed using SPSS (version 29.0.0.0 [241], IBM, Armonk, NY, USA). Mann–Whitney U test was used to compare continuous variables (e.g., age) between the two groups at the first visit, and Fisher’s exact test was used to compare categorical variables (e.g., sex). Pearson’s correlation analysis was used to evaluate the relationships between CCT and BCVA, as well as between the L/C ratio and BCVA at baseline. The Friedman test was used to compare continuous variables (e.g., CCT, L/C ratio) among the three periods (first visit, after the first injection, and after the third injection), followed by post hoc pairwise comparisons using the Bonferroni correction. Repeated Measures ANOVA was used to compare changes through time in the L/C ratios between the MNVs in the faricimab group. In performing Repeated Measures ANOVA, we confirmed the normal distribution with the Kolmogorov–Smirnov test and the sphericity with the Mauchly test. The threshold for statistical significance was set at *p* < 0.05.

## 3. Results

A total of 28 patients (28 eyes) with nAMD were included in the study. Seventeen eyes (61%) were diagnosed with Type 1 MNV, including 9 eyes with polyps. Eleven eyes (39%) were diagnosed with Type 2 MNV. The faricimab and ranibizumab groups consisted of 13 patients (13 eyes) and 15 patients (15 eyes), respectively. All patient backgrounds are shown in Table 1.

The median age was 77 years. In total, 20 patients (71%) were male and 8 patients (29%) were female. Ten patients (36%) had a history of cardiovascular disease. Type 1 MNV was observed in 17 patients (61%). SRF was observed in 27 out of 28 patients (96%) and IRF was observed in 11 out of 28 patients (39%). There was no significant correlation between CCT and BCVA in the total patients before treatment (r = 0.073, *p* = 0.71). Similarly, no significant correlation was observed between the L/C ratio and BCVA (r = 0.098, *p* = 0.62). A comparison between drugs at the initial visit is shown in Table 2.

The median age was significantly younger in the faricimab group, but there was no difference in gender between the two groups (*p* = 0.021 and *p* > 0.999, respectively). There were also no significant differences between the two groups in the type of MNV, the incidence of polyps, or the presence of drusen. The incidence of IRF was significantly higher in the ranibizumab group, while there was no significant difference in the presence of SRF between the two groups (*p* = 0.024 and *p* > 0.999, respectively). Initial BCVA was significantly worse in the ranibizumab group (*p* = 0.012). There were no significant differences between the two groups in CCT (*p* = 0.93), TCA (*p* = 0.79), LCA (*p* = 0.68), or SCA (*p* = 0.93). In contrast, the L/C ratio was significantly higher in the faricimab group (*p* = 0.029). Representative cases showing the treatment course for the faricimab and ranibizumab groups are presented in the OCT images and their corresponding binarized images (Figure 2 and Figure 3).

The clinical changes of BCVA after anti-VEGF agents are shown in Figure 4.

There was no significant difference in BCVA for both faricimab and ranibizumab during treatment (*p* = 0.12, 0.94, respectively). After the third injection, dry macula was achieved in 62% (8 patients) of the faricimab group and in 60% (9 patients) of the ranibizumab group. The clinical changes of CCT after anti-VEGF agents are shown in Figure 5.

In the faricimab group, there were no significant changes in CCT among the three vitreous injections (*p* = 0.23). In contrast, CCT significantly decreased during treatment in the ranibizumab group (*p* = 0.006). Post hoc analysis with Bonferroni correction revealed a significant reduction in CCT after the first injection compared to baseline (*p* = 0.015). However, no significant difference was observed in CCT after the third injection compared to baseline (*p* = 0.32). The clinical changes of the L/C ratio after anti-VEGF agents are shown in Figure 6.

In the faricimab group, the L/C ratio showed a significant change during treatment (*p* = 0.018). Post hoc analysis with Bonferroni correction revealed a significant reduction in the L/C ratio after the third injection compared to baseline (*p* = 0.010). Although no significant difference was observed in the L/C ratio after the first injection compared to baseline (*p* = 0.57), the L/C ratio after the third injection showed a significant reduction compared to after the first injection (*p* = 0.032). On the contrary, no significant changes were observed in the ranibizumab group (*p* = 0.34). Changes in the L/C ratio for each type of MNV during treatment in the faricimab group are shown in Figure 7.

The analysis revealed a significant main effect of time (F (2,22) = 10.38, *p* < 0.001), indicating that the L/C ratio changed significantly throughout treatment. Although the main effect of MNV group was not statistically significant (F (1,11) = 3.76, *p* = 0.079), a significant interaction between MNV group and time was observed (F (2,22) = 4.97, *p* = 0.017), meaning that the pattern of change in the L/C ratio differed significantly between type 1 and type 2 MNV groups. Post hoc analysis with Bonferroni correction showed significant differences between baseline and after the third injection (*p* = 0.004), and between after the first injection and after the third injection (*p* = 0.045), indicating a significant reduction in the L/C ratio after three injections.

## 4. Discussion

This pilot study evaluated OCT-based structural changes in the choroid during intravitreal injections of anti-VEGF drugs. As an exploratory analysis, we observed significant differences between the faricimab and ranibizumab groups in several baseline characteristics.

Although TCA, LCA, and SCA did not differ significantly between the two treatment groups at baseline, the L/C ratio was significantly higher in the faricimab group. These findings suggest that proportional alterations in the luminal and stromal components, rather than changes in the absolute choroidal area, may be more sensitive in structural differences. Previous studies have suggested that an elevated L/C ratio may predict exudative activity [12,13,14]. Therefore, it is plausible that faricimab was chosen for patients with more pronounced exudative changes. Beyond baseline differences, notable differences in treatment response were also observed between the two agents. The differences could be attributed to the different pharmacological profiles: i.e., faricimab is a bispecific antibody targeting both VEGF-A and Ang-2, while ranibizumab is a Fab fragment that targets only VEGF-A [18].

The median age was significantly higher in the ranibizumab group. The older patients might have been preferentially treated with ranibizumab due to physicians’ consideration of its lower systemic exposure. Baseline BCVA was significantly worse in the ranibizumab group, possibly due to age-related cataracts. Additionally, the incidence of IRF was higher in the ranibizumab group, suggesting that the presence of IRF may have influenced physicians’ decision to select this agent.

Although the suppression of VEGF has been emphasized in inhibiting the progression of nAMD, new mechanisms of angiogenesis in nAMD have been emerging. Angiogenesis occurs when pericytes are shed from normal choroidal vessels, leading to the dilation of existing vessels and the formation of abnormal new vessels [19,20]. The angiopoietin 1 (Ang-1)/tyrosine kinase with immunoglobulin and epidermal growth factor homology domains 2 (Tie-2) signaling pathway has an inhibitory effect on angiogenesis [21,22]. However, Ang-2, a Tie-2 antagonist, destabilizes blood vessels by binding to Tie-2 receptors on endothelial cells [23,24]. Ang-2 production is increased in eyes with nAMD, which inhibits the activation of Tie-2 as an adhesive effect on vascular endothelia [25]. While anti-VEGF action causes regression of immature neovascular vessels, some vessels expand without regression [26,27].

Faricimab, hording an anti-Ang-2 effect as well, activates normal Ang-1/Tie-2 signaling, which results in neovascular maturation and stabilizing endothelial cell adhesion [28]. Previous studies have shown that faricimab inhibits both VEGF and Ang-2, thus contributing to vascular stability and reduction of inflammation [5,29]. In our study, the L/C ratio significantly decreased in the faricimab group after three injections. This result suggests that the anti-Ang-2 effect in the faricimab group may not only affect the MNV but also the entire choroidal vasculature, causing narrowing of the choroidal lumen. Although a previous report has shown reduced blood flow signals in the choriocapillaris after faricimab treatment [30], comparisons of pre- and post-treatment entire choroidal vasculatures have been lacking. Thus, this pilot study provides preliminary insights into the potential effects of faricimab on choroidal vasculatures.

A previous study reported a decrease in the L/C ratio after aflibercept treatment in nAMD patients [14]. Notably, there have been reports demonstrating faster anatomic improvements with faricimab than with aflibercept [31]. The anti-Ang-2 effect may affect long-term structural changes in the choroidal vasculature. Moreover, only CCT significantly decreased after the first injection in the ranibizumab group. This result suggests that while ranibizumab temporarily causes choroidal thinning, its effect on the choroidal luminal system may be limited. Although ranibizumab biosimilar has also been used in recent years [32], ranibizumab targets only VEGF-A; therefore, its overall effect on the choroidal vasculature may have been less marked.

We further analyzed the changes in the L/C ratio according to MNV type in the faricimab group. The interaction between MNV type and time was significant, demonstrating that a significant reduction in the L/C ratio was observed only in eyes with type 1 MNV, but not in type 2 MNV. This suggests that the treatment effect on choroidal vasculatures may be more pronounced in type 1 MNV. Due to its location, type 1 MNV is more directly associated with choroidal vessels, and the L/C ratio in Type 1 MNV may have been more sensitive to structural changes or blood flow changes induced by the faricimab treatment. While a recent study has reported that the L/C ratio is often lower in Type 1 MNV compared to Type 2 MNV [33], this pilot study showed no difference in the L/C ratio between Type 1 and Type 2 MNV before the treatment. Previous reports have shown that faricimab has a strong effect in reducing the size of exudate fluids in Type 1 lesions [34,35]. The rapid absorption of exudate fluid may alter the choroidal vascular structures, resulting in the reduction of the L/C ratio. Our preliminary findings suggest that the L/C ratio has potential as a biomarker for monitoring choroidal vascular remodeling and treatment efficacy in type 1 MNV.

This study has several limitations. First, as a retrospective study, the choice of treatment agents was determined by individual physicians, potentially influenced by patient-specific factors such as comorbidities and socioeconomic status. This study initially compared the differences between faricimab and ranibizumab. Therefore, future comparisons with other drugs, such as aflibercept, aflibercept 8 mg, brolucizumab, and ranibizumab biosimilar, should be made. Second, only patients with access to high-quality OCT b-scan images were included. Consequently, severely affected individuals, such as those with subretinal hemorrhages or large pigment epithelial detachments, were excluded due to image blurring. Future studies should aim to develop imaging protocols that allow reliable analysis even in patients with advanced disease. Third, this was a single-center study with a relatively small sample size for each treatment group. However, as a pilot study, our primary goal was to generate hypotheses and provide exploratory evidence rather than establish definitive conclusions. Future multicenter prospective studies with larger cohorts and standardized imaging protocols are warranted to validate these preliminary observations.

## 5. Conclusions

In this pilot study, our results suggest that faricimab may exert anti-VEGF and Ang-2 effects predominantly on type 1 MNV, leading to potential vascular constriction. While these findings offer preliminary guidance for treatment selection in patients with different MNV subtypes, larger studies are needed to establish their clinical significance. In addition, longer-term studies are warranted to determine whether the effects of faricimab on choroidal vascular structure are sustained rather than transient.

## Figures and Tables

**Figure 1 jcm-14-07257-f001:**
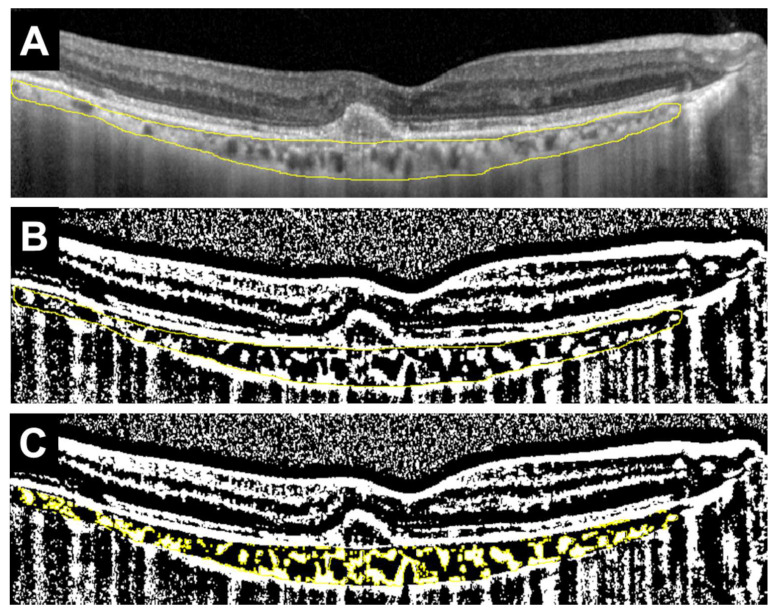
Example of measuring the ratio of luminal area to choroidal area (L/C ratio) in a horizontal OCT image. (**A**) Select the total choroidal area (TCA) in the arcade (area surrounded by the yellow line). (**B**) Binarize the OCT image using the Niblack method (black area and white area). White areas are defined as stromal choroidal area (SCA), whereas black areas are defined as luminal choroidal area (LCA). (**C**) Identify SCA (shown in yellow), and calculate SCA and TCA. LCA is calculated by deducting SCA from TCA, and the L/C ratio is defined as the ratio of LCA to TCA (%).

**Figure 2 jcm-14-07257-f002:**
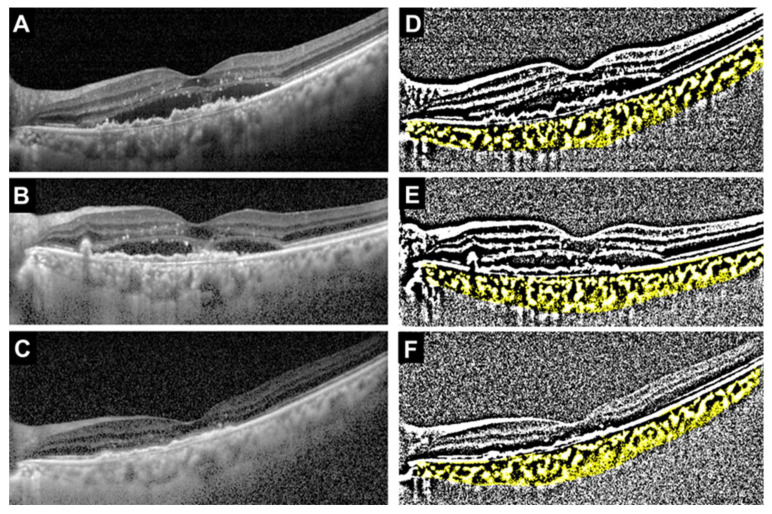
Representative case treated with faricimab (69-year-old female). (**A**) Baseline OCT image showing type 1 MNV with polyps and subretinal fluid (SRF). (**B**) OCT image after the first intravitreal injection showing a reduction in SRF. (**C**) OCT image after the third injection demonstrating complete resolution of SRF. (**D**–**F**) Corresponding binarized OCT images at baseline, after the first injection, and after the third injection, respectively. The total choroidal area (TCA), luminal choroidal area (LCA), stromal choroidal area (SCA, shown in yellow), and L/C ratio at each time point were as follows: Baseline: TCA = 2.54 mm^2^, LCA = 1.67 mm^2^, SCA = 0.88 mm^2^, L/C ratio = 65.5%. After the first injection: TCA = 1.76 mm^2^, LCA = 1.05 mm^2^, SCA = 0.71 mm^2^, L/C ratio = 59.4%. After the third injection: TCA = 1.86 mm^2^, LCA = 1.09 mm^2^, SCA = 0.77 mm^2^, L/C ratio = 58.5%.

**Figure 3 jcm-14-07257-f003:**
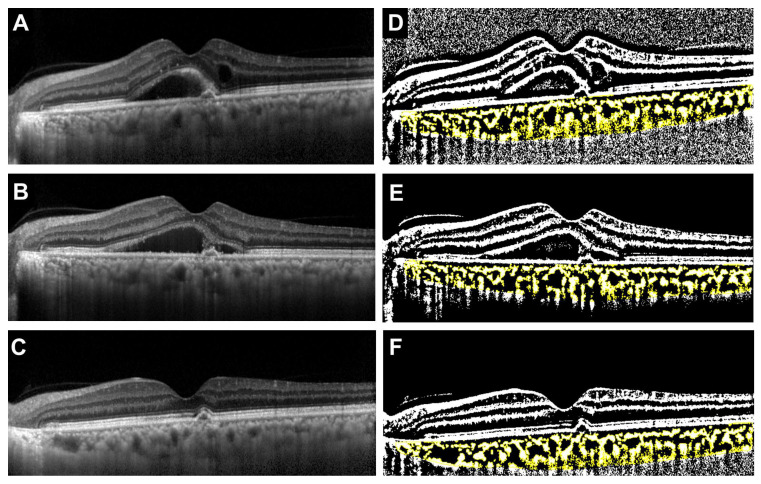
Representative case treated with ranibizumab (68-year-old male). (**A**) Baseline OCT image showing type 1 MNV with polyps, intraretinal fluid (IRF), and SRF. (**B**) OCT image after the first intravitreal injection demonstrating regression of IRF, while SRF remained unchanged. (**C**) OCT image after the third injection showing complete resolution of SRF. (**D**–**F**) Corresponding binarized OCT images at baseline, after the first injection, and after the third injection, respectively. The total choroidal area (TCA), luminal choroidal area (LCA), stromal choroidal area (SCA, shown in yellow), and L/C ratio at each time point were as follows: Baseline: TCA = 1.95 mm^2^, LCA = 1.21 mm^2^, SCA = 0.73 mm^2^, L/C ratio = 62.4%. After the first injection: TCA = 2.01 mm^2^, LCA = 1.28 mm^2^, SCA = 0.72 mm^2^, L/C ratio = 63.9%. After the third injection: TCA = 1.64 mm^2^, LCA = 1.06 mm^2^, SCA = 0.58 mm^2^, L/C ratio = 64.6%.

**Figure 4 jcm-14-07257-f004:**
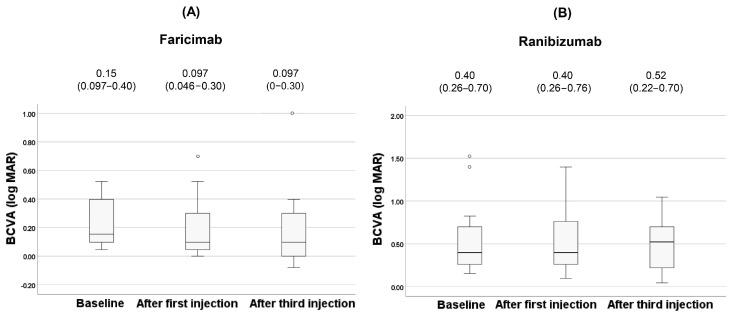
Changes of BCVA during the treatment in the faricimab group (**A**) and in the ranibizumab group (**B**). Each of the values in the graph is the median (interquartile range, IQR). White circles represent statistical outliers.

**Figure 5 jcm-14-07257-f005:**
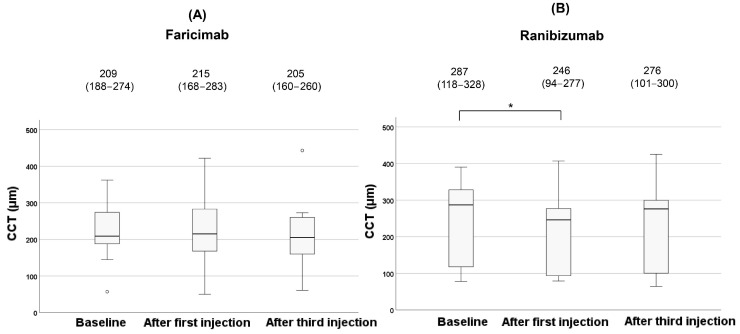
Changes of CCT during the treatment in the faricimab group (**A**) and in the ranibizumab group (**B**). Each of the values in the graph is the median (interquartile range, IQR). Statistical significance was determined at *p* < 0.05, indicated by *. White circles represent statistical outliers.

**Figure 6 jcm-14-07257-f006:**
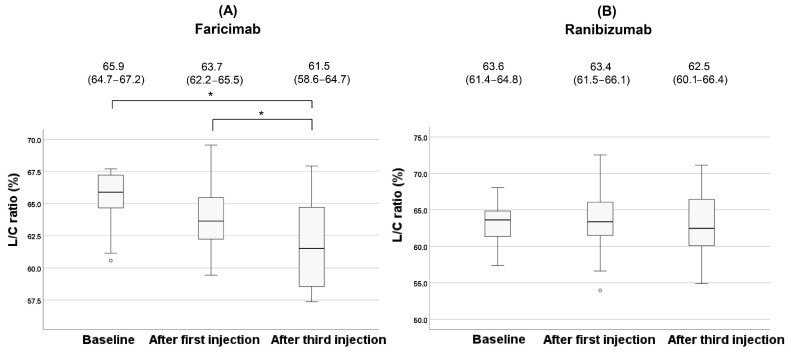
Changes of the L/C ratio during the treatment in the faricimab group (**A**) and in the ranibizumab group (**B**). Each of the values in the graph is the median (interquartile range, IQR). Statistical significance was determined at *p* < 0.05, indicated by *. White circles represent statistical outliers.

**Figure 7 jcm-14-07257-f007:**
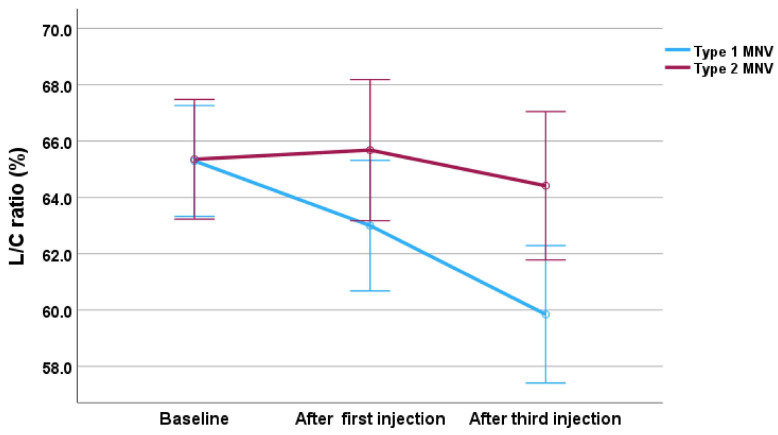
Changes in the L/C ratio during the treatment, divided by MNV type in the faricimab group. The blue line represents type 1 MNV, and the red line represents type 2 MNV. Error bars represent the 95% confidence interval.

**Table 1 jcm-14-07257-t001:** Characteristics of total patients (*n* = 28).

Age (IQR)	77 (68–82)
Sex (%)	male: 20 (71) female: 8(29)
Cardiovascular disease (%)	10 (36)
Type of MNV	
Type 1 (%)	17 (61)
Polyps (%, in Type 1 MNV)	9 (53)
Type 2 (%)	11 (39)
drusen (%)	21 (75)
IRF (%)	11 (39)
SRF (%)	27 (96)
BCVA (logMAR, IQR)	0.30 (0.15–0.52)
CCT (μm, IQR)	230 (142–302)
TCA (mm^2^, IQR)	1.33 (1.03–1.80)
LCA (mm^2^, IQR)	0.84 (0.63–1.16)
SCA (mm^2^, IQR)	0.47 (0.37–0.64)
L/C ratio (%, IQR)	64.7 (62.4–66.1)

IQR: interquartile range, MNV: macular neovascularization, IRF: intraretinal fluid, SRF: subretinal fluid, BCVA: best-corrected visual acuity, CCT: central choroidal thickness, TCA: total choroidal area, SCA: stromal choroidal area, LCA: luminal choroidal area, L/C ratio: the ratio of luminal area to choroidal area.

**Table 2 jcm-14-07257-t002:** Characteristics of patients at initial visit classified by anti-VEGF agents.

	Faricimab *(n* = 13)	Ranibizumab (*n* = 15)	*p*
Age (IQR)	69 (65–78)	81 (75–83)	0.021 ^†^
Sex (%)	male: 9 (69) female: 4 (31)	male: 11 (73) female: 4 (27)	>0.999 ^‡^
Cardiovascular disease (%)	3 (23)	7 (47)	0.25 ^‡^
Type of MNV			
Type 1 (%)	7 (54)	10 (67)	0.70 ^‡^
Polyps (%, in Type 1 MNV)	5 (71)	4 (40)	0.34 ^‡^
Type 2 (%)	6 (46)	5 (33)	-
drusen (%)	8 (62)	13 (87)	0.20 ^‡^
IRF (%)	2 (15)	9 (60)	0.024 ^‡^
SRF (%)	13 (100)	14 (93)	>0.999 ^‡^
BCVA (logMAR, IQR)	0.15 (0.097–0.40)	0.40(0.26–0.70)	0.012 ^†^
CCT (μm, IQR)	209 (188–274)	287 (118–328)	0.93 ^†^
TCA (mm^2^, IQR)	1.30 (1.19–1.76)	1.42 (0.95–1.82)	0.79 ^†^
LCA (mm^2^, IQR)	0.83 (0.80–1.14)	0.91 (0.58–1.16)	0.68 ^†^
SCA (mm^2^, IQR)	0.44 (0.39–0.63)	0.50 (0.35–0.64)	0.93 ^†^
L/C ratio (%, IQR)	65.9 (64.7–67.2)	63.6 (61.4–64.8)	0.029 ^†^

IQR: interquartile range, IRF: intraretinal fluid, SRF: subretinal fluid, BCVA: best-corrected visual acuity, CCT: central choroidal thickness, TCA: total choroidal area, SCA: stromal choroidal area, LCA: luminal choroidal area, L/C ratio: the ratio of luminal area to choroidal area. ^†^ Mann–Whitney U test. ^‡^ Fisher’s exact test.

## Data Availability

The datasets generated or analyzed in the current study are available from the corresponding author upon reasonable request.

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
