# Peer review of "A Pilot Study on Structural Changes of Choroidal Vasculature Following Intravitreal Anti-VEGF Injection in Neovascular Age-Related Macular Degeneration: Faricimab vs Ranibizumab"

_jcm, 2025, doi:10.3390/jcm14207257_

Round 1

Reviewer 1 Report

Comments and Suggestions for Authors

Authors Takeyuki Nishiyama in the article entitled ‘A pilot study on structural changes of choroidal vasculature following intravitreal faricimab injection in neovascular age-related macular degeneration’ have performed a satisfactory study to show the positive impact of faricimab on nAMD patients. The article has been written well including the results generation by applying appropriate statistical analysis. However, some information that may be needed to be incorporated in the article as listed below:

  1. As illustrated in Table 1 and Table 2, was there any differences noted between in baseline vs after treatment between two groups in the MNV type? Can a table be inserted in section 3 to show summarize this result.
  2. As detailed in the article, the injection regimen was conducted between Jan-Feb 2025 and accordingly the ophthalmic examinations were conducted to obtain the results. What is your opinion on following up on an extended examination schedule to ensure the treatment of faricimab is staying on the long-term scale but not transient. (May be a comment on this can inserted in section 5, as a future step to consider)
  3. Will the CCT or L/C ratio impact the visual acuity? Especially its impact on the intra ocular pressure after three intra-vitreal injections on the nAMD patients can be mentioned in section 4.

Author Response

Reviewer1

Authors Takeyuki Nishiyama in the article entitled ‘A pilot study on structural changes of choroidal vasculature following intravitreal faricimab injection in neovascular age-related macular degeneration’ have performed a satisfactory study to show the positive impact of faricimab on nAMD patients. The article has been written well including the results generation by applying appropriate statistical analysis. However, some information that may be needed to be incorporated in the article as listed below:

1. As illustrated in Table 1 and Table 2, was there any differences noted between in baseline vs after treatment between two groups in the MNV type? Can a table be inserted in section 3 to show summarize this result.

Response: Thank you very much for your valuable comment. As you correctly pointed out, previous studies have reported that MNV types may change during the course of treatment. However, because our study involved a short observation period limited to the first three loading injections, no changes in MNV type were observed in our data. For this reason, we believe that adding table at this stage would not be appropriate.

We sincerely appreciate your insightful suggestion, and we will consider this point in future studies with longer follow-up periods.

2. As detailed in the article, the injection regimen was conducted between Jan-Feb 2025 and accordingly the ophthalmic examinations were conducted to obtain the results. What is your opinion on following up on an extended examination schedule to ensure the treatment of faricimab is staying on the long-term scale but not transient. (May be a comment on this can inserted in section 5, as a future step to consider)

Response: Yes, neovascular AMD is a lifelong disease, and most patients require continuous anti-VEGF treatment. Therefore, it is important to investigate the long-term effects of faricimab to determine whether its influence on the choroidal vascular structure is sustained rather than transient.

As you suggested, we added a following sentence in section 5 (Conclusions: Page 12 Line 311-313): In addition, longer-term studies are warranted to determine whether the effects of faricimab on choroidal vascular structure are sustained rather than transient.

3. Will the CCT or L/C ratio impact the visual acuity? Especially its impact on the intra ocular pressure after three intra-vitreal injections on the nAMD patients can be mentioned in section 4.

Response: Thank you very much for your thoughtful comment. In our study, neither CCT nor the L/C ratio showed any association with visual acuity. As you suggested, we added the following sentences to the Results section (Page 5 Line 135-137): There was no significant correlation between CCT and BCVA in all patients before treatment (r = 0.073, P = 0.71). Similarly, no significant correlation was observed between the L/C ratio and BCVA (r = 0.098, P = 0.62).

Regarding intraocular pressure, no significant changes were observed after the three intravitreal injections; therefore, we did not include this parameter in the manuscript. We appreciate your suggestion and will further investigate these parameters in future studies.

Reviewer 2 Report

Comments and Suggestions for Authors

Dear Authors:

Please find the attached comments/suggestions.

Best regards,

Maria.

Author Response

Reviewer2

Comments In this pilot study the authors retrospectively explored structural choroidal vascular changes by optical coherence tomography in patients with neovascular age-related macular degeneration (nAMD) treated with faricimab and ranibizumab. This is a well-written paper. However, there are some comments to be addressed by the authors. Please find the following comments:

1. According to the authors the title of this study is: “A pilot study on structural changes of choroidal vasculature following intravitreal faricimab injection in neovascular age-related macular degeneration”. However, the title does not include ranibizumab.

Response: Thank you very much for your comment. Following your suggestion, we revised the title as follows: A pilot study on structural changes of choroidal vasculature following intravitreal anti-VEGF injection in neovascular age-related macular degeneration: Faricimab vs. Ranibizumab.

 2. What was the diagnosis of these 28 patients? Please add this information.

Response: According to the reviewer, we revised and added the following sentences to the Results section (Page 5 Line 125-127): A total of 28 patients (28 eyes) with nAMD were included in the study. Seventeen eyes (61%) were diagnosed with Type 1 MNV, including 9 eyes with polyps. Eleven eyes (39%) were diagnosed with Type 2 MNV.

3. The interval for measuring structural choroidal vascular changes and other parameters after intravitreal injection of anti-VEGF drugs varies depending on the study. What was the timeframe in this study?

Response: Thank you for pointing this out. Measurements were performed two weeks after the first injection and three months after the first injection (i.e., one month after the third injection). Following your suggestion, we have added this information regarding the measurement intervals to the revised manuscript in the Materials and Methods section (Page 3 Line 72-75) as follows: By reviewing the medical records, this study explored best-corrected visual acuity (BCVA) and OCT images for patients with nAMD before treatment, 2 weeks after the first anti-VEGF vitreous injection, and 1 month after the third injection (3 months after the first injection).

4. Please add the % of eyes with Type 2 MNV to the Abstract Section.

Response: According to the reviewer, we added the % of the eyes with Type 2 MNV to the abstract section.

5. Introduction Section: the bibliography is adequate. However, information about macular neovascularization classification and subretinal fluid as indicator for evaluating the disease activity of MNV is missing.

Response: Thank you very much for your valuable comment. Following your suggestion, we have added reference 2 (Mathis et al., 2023) and revised the Introduction to include the following sentences (Page 1-2 Line 36-42): Macular neovascularization (MNV) is generally classified into Type 1 and Type 2 based on the location of the neovascular complex relative to the retinal pigment epithelium (RPE) [2]. In Type 1 MNV, the neovascular complex is located beneath the RPE, whereas in Type 2 MNV, it is located above the RPE in the subretinal space. Visual impairment in patients with MNV is primarily caused by fluid leakage and hemorrhage, which can ultimately lead to fibrous scarring [3]. Subretinal fluid (SRF) is one of the key indicators used to evaluate disease activity in MNV, reflecting ongoing leakage from the neovascular complex [4].

 6. Line 82: why did the authors not binarize both horizontal and vertical OCT scans?

Response: Thank you very much for your comment. In our previous study (Hirai H, Yamashita M, Ijuin N, et al. J Clin Med. 2024;13:1383) [13], we analyzed both horizontal and vertical OCT scans. The results showed that only the L/C ratio obtained from horizontal OCT scans was significantly associated with subretinal fluid (SRF). Therefore, in the present study, we used horizontal OCT images only.

7. In page 3, Figure 1-A: please specify the entire choroidal area within the arcade using color.

Response: According to the reviewer, we revised Figure 1A and specified the entire choroidal area within the arcade.

8. Please include the % of females in the revised manuscript.

Response: According to the reviewer, we added the % of females in Tables 1, 2, and Page 5 Line 132-133.

9. In order to improve the manuscript, the authors need to include the following Figures:

9a)-An enhanced depth imaging SD-OCT from representative eyes at baseline and after the first and third intravitreal injections with Faricimab and Ranibizumab to confirm the presence of IRF, SRF, Type of MNV, polyps, CCT.

Thank you for your valuable comment. In accordance with your suggestion, we have added representative cases: a Faricimab-treated case (Figure 2 A-C) and a Ranibizumab-treated case (Figure 3 A-C) in the Results section.

9b)-An OCT scan binarized image from representative eyes showing the L/C ratio changes before treatment and after the first and third intravitreal injections with anti-VEGF agents. Please add the values of LCA, SCA, TCA, and L/C ratio for each group.

Response: Thank you for your valuable comment. In accordance with your suggestion, we have added representative binarized OCT images from the faricimab group (Figure 2 D-F) and the ranibizumab group (Figure 3 D-F) to the Results section, showing L/C ratio changes before treatment, after the first injection, and after the third injection. The corresponding values of TCA, LCA, SCA, and the L/C ratio for each case are included in the figure legends.

10. In line 122: please add IRF %.

Response: According to the reviewer, we added the IRF % in the revised manuscript.

11. In Figure 2, 3, and 4: please increase the size of X and Y axis labels.

Response: According to the reviewer, we fixed Figure 2, 3, and 4.

12. In line 225: the number 13 after patients. should be a reference. Please correct this.

Response: According to the reviewer, we corrected the revised manuscript.

13. I agree with the authors about the limitations of this study.

Response: Thank you very much for your feedback. As mentioned in the manuscript, we acknowledge the limitations of this pilot study and plan to conduct further research with a larger sample size in the future to confirm and expand upon our findings.

Reviewer 3 Report

Comments and Suggestions for Authors

Comments for Manuscript "A pilot study on structural changes of choroidal vasculature following intravitreal faricimab injection in neovascular age-related macular degeneration" submitted to  journal of clinical medicine

This article, submitted by Takeyuki Nishiyama et al. and titled “A pilot study on structural changes of choroidal vasculature following intravitreal faricimab injection in neovascular age-related macular degeneration”, retrospectively evaluated the outcomes of two anti-VEGF agents, faricimab and ranibizumab, with respect to best-corrected visual acuity (BCVA) and structural markers such as central choroidal thickness and the luminal-to-choroidal area ratio (L/C ratio) in 28 treatment-naïve nAMD patients, based on OCT images. In addition, the study compared the effect of faricimab on changes in the L/C ratio between Type 1 and Type 2 MVN patients.

To the best of my knowledge, this is the first report suggesting that faricimab may contribute to regression of nAMD by altering the structure of the entire choroidal vasculature. The authors also propose a potential clinical implication, indicating that faricimab may be more effective in Type 1 MVN. However, there existed some issues that needed to be addressed. With the aim to help the authors to improve the manuscript, I have some comments for their consideration:

  1. To ensure all results mentioned in the article are covered, best-corrected visual acuity (BCVA) outcomes should be included in the Results section of the Abstract.
  2. Line 86:It is written that “Select the entire choroidal zone in the arcade.”However, the specified area is not outlined in Image A of Figure 1. It is recommended to mark the specified area in Image A to illustrate the measurement method more clearly.
  3. Since the patient characteristics are clearly presented in Table 2, Table 1 appears redundant, and it is recommended to delete it.
  4. To clarify whether the anti-Ang-2 or anti-VEGF function predominantly affects the change in L/C in the two types of MVN, it would be helpful if the authors could provide data comparing the effects of faricimab and ranibizumab on Type 1 and Type 2 MVN.
  5. The title and objective of the manuscript are not fully aligned. The title, “A pilot study on structural changes of choroidal vasculature following intravitreal faricimab injection in neovascular age-related macular degeneration,” implies that the study focuses exclusively on faricimab. However, the stated objective describes the evaluation of choroidal vascular changes in patients treated with both faricimab and ranibizumab. This discrepancy may confuse readers about the true scope of the study.

Author Response

Reviewer3

Comments for Manuscript "A pilot study on structural changes of choroidal vasculature following intravitreal faricimab injection in neovascular age-related macular degeneration" submitted to  “journal of clinical medicine”

This article, submitted by Takeyuki Nishiyama et al. and titled “A pilot study on structural changes of choroidal vasculature following intravitreal faricimab injection in neovascular age-related macular degeneration”, retrospectively evaluated the outcomes of two anti-VEGF agents, faricimab and ranibizumab, with respect to best-corrected visual acuity (BCVA) and structural markers such as central choroidal thickness and the luminal-to-choroidal area ratio (L/C ratio) in 28 treatment-naïve nAMD patients, based on OCT images. In addition, the study compared the effect of faricimab on changes in the L/C ratio between Type 1 and Type 2 MVN patients.To the best of my knowledge, this is the first report suggesting that faricimab may contribute to regression of nAMD by altering the structure of the entire choroidal vasculature. The authors also propose a potential clinical implication, indicating that faricimab may be more effective in Type 1 MVN. However, there existed some issues that needed to be addressed. With the aim to help the authors to improve the manuscript, I have some comments for their consideration:

1. To ensure all results mentioned in the article are covered, best-corrected visual acuity (BCVA) outcomes should be included in the Results section of the Abstract.

Response: According to the Reviewer, we added the BCVA outcomes in the abstract section as follows: There was no significant difference in best corrected visual acuity (BCVA) for both faricimab and ranibizumab during treatment (P = 0.12, 0.94, respectively).

2. Line 86: It is written that “Select the entire choroidal zone in the arcade.” However, the specified area is not outlined in Image A of Figure 1. It is recommended to mark the specified area in Image A to illustrate the measurement method more clearly.

Response: According to the reviewer, we revised Figure 1A and specified the entire choroidal area within the arcade.

3. Since the patient characteristics are clearly presented in Table 2, Table 1 appears redundant, and it is recommended to delete it.

Response: Thank you very much for your comment. While we understand that Table 2 presents the baseline characteristics of each treatment group, we believe that Table 1 is still necessary as it summarizes the overall characteristics of all patients in the study. Therefore, we would like to keep Table 1 to provide a clear overview of the total patient population.

4. To clarify whether the anti-Ang-2 or anti-VEGF function predominantly affects the change in L/C in the two types of MVN, it would be helpful if the authors could provide data comparing the effects of faricimab and ranibizumab on Type 1 and Type 2 MVN.

Response: Thank you very much for your comment. In our study, no significant differences were observed in the overall ranibizumab-treated group. Therefore, we considered subgroup analysis by MNV type for ranibizumab to be unnecessary. Nevertheless, we appreciate your suggestion and will consider this approach in future studies to further clarify the effects of anti-VEGF and anti-Ang-2 therapies on different MNV types.

5. The title and objective of the manuscript are not fully aligned. The title, “A pilot study on structural changes of choroidal vasculature following intravitreal faricimab injection in neovascular age-related macular degeneration,” implies that the study focuses exclusively on faricimab. However, the stated objective describes the evaluation of choroidal vascular changes in patients treated with both faricimab and ranibizumab. This discrepancy may confuse readers about the true scope of the study.

Response: Thank you for pointing this out. Following your suggestion, we revised the title as follows: A pilot study on structural changes of choroidal vasculature following intravitreal anti-VEGF injection in neovascular age-related macular degeneration: Faricimab vs. Ranibizumab.